# Metabolic Syndrome Prevalence in Women with Gestational Diabetes Mellitus in the Second Trimester of Gravidity

**DOI:** 10.3390/jcm13051260

**Published:** 2024-02-22

**Authors:** Vendula Bartáková, Katarína Chalásová, Lukáš Pácal, Veronika Ťápalová, Jan Máchal, Petr Janků, Kateřina Kaňková

**Affiliations:** 1Department of Pathophysiology, Faculty of Medicine, Masaryk University, 62500 Brno, Czech Republicpaci@med.muni.cz (L.P.); jmachal@med.muni.cz (J.M.); kankov@med.muni.cz (K.K.); 2Department of Obstetrics and Gynaecology, University Hospital Brno, 66263 Brno, Czech Republic; tapalova.veronika@fnbrno.cz (V.Ť.); janku.petr@fnbrno.cz (P.J.)

**Keywords:** gestational diabetes mellitus, metabolic syndrome, oral glucose tolerance test, postpartum glucose intolerance, pregnancy

## Abstract

**Background**: Women with gestational diabetes (GDM) have an increased risk of metabolic syndrome (MS) after delivery. MS could precede gravidity. The aims of this study were (i) to detect the prevalence of MS in women at the time of GDM diagnosis, (ii) to detect the prevalence of MS in the subgroup of GDM patients with any form of impaired glucose tolerance after delivery (PGI), and (iii) to determine whether GDM women with MS have a higher risk of peripartal adverse outcomes. **Methods***:* A cross-sectional observational study comprised *n* = 455 women with GDM. International Diabetes Federation (IDF) criteria for MS definition were modified to the pregnancy situation. **Results***:* MS was detected in 22.6% of GDM patients in those with PGI 40%. The presence of MS in GDM patients was associated with two peripartal outcomes: higher incidence of pathologic Apgar score and macrosomia (*p* = 0.01 resp. *p* = 0.0004, chi-square). **Conclusions***:* The presence of MS in GDM patients is a statistically significant risk factor (*p* = 0.04 chi-square) for PGI. A strong clinical implication of our findings might be to include MS diagnostics within GDM screening using modified MS criteria in the second trimester of pregnancy.

## 1. Introduction

Pregnancy can be complicated by several forms of diabetes. Diabetes could precede pregnancy—as a pre-existing type 1 diabetes mellitus or type 2 diabetes and less often as Maturity Onset Diabetes of the Young (MODY). However, most commonly, impaired glucose tolerance manifests for the first time during pregnancy (in either the first or second trimester) as gestational diabetes mellitus (GDM). GDM is traditionally defined as any degree of glucose intolerance first recognised during pregnancy, regardless of the degree of hyperglycaemia [1]. According to the American Diabetes Association (ADA) “Standards of Medical Care in Diabetes” 2022, GDM is defined as diabetes diagnosed in the second or third trimester of pregnancy that was not clearly overt diabetes prior to gestation [2]. Furthermore, by some definitions, GDM should normalise after pregnancy; however, the true regression rate is not known since participation in postpartum screening is in general low (only about 50% of women diagnosed with GDM during pregnancy attend oGTT after delivery) [3]. Moreover, published data document a significantly increased risk of type 2 diabetes and cardiovascular diseases (CVDs) after delivery or anytime in later life [4].

During a healthy pregnancy, the mother’s body undergoes a series of physiological changes. One important metabolic adaptation is a change in insulin sensitivity. During early gestation, insulin sensitivity increases, promoting the uptake of glucose into adipose stores in anticipation of the increased energy demands of later pregnancy. However, as pregnancy progresses, a surge in maternal and placental hormones, including oestrogen, progesterone, leptin, cortisol, placental lactogen, and placental growth hormone collectively promote a state of insulin resistance. Pregnant women compensate for these changes by hypertrophy and hyperplasia of pancreatic beta cells, as well as increased glucose-stimulated insulin secretion with its subsequent desirable anabolic activity but without hyperglycaemia [5].

The above-described physiological metabolic adaptations to pregnancy do not adequately occur in all pregnancies, resulting in GDM marked by hyperglycaemia typically manifested in mid-gestation. The established risk factors for GDM include overweight/obesity, a westernised diet and micronutrient deficiencies, advanced maternal age, and a family history of insulin resistance and/or diabetes [6]. Published data show a significant association between GDM and obesity and other components of metabolic syndrome (MS) [7]. MS is generally defined as a combination of several disturbances comprising insulin resistance or frank diabetes, higher body fat (higher waist circumference or body mass index, BMI), dyslipidaemia, and hypertension [6]; see Table 1 for commonly used MS definitions. The prevalence of MS and GDM is rising worldwide, and both phenotypes of impaired glucose tolerance share certain common features and risks, mainly increased body fat, the effect of age, and insulin secretory defects. Definitions of both MS and GDM are not unified worldwide and have changed over time. We may assume that a subset of GDM cases, and especially those with MS, represent in fact type 2 diabetes first manifested in pregnancy since pregnancy-related metabolic changes can unmask susceptibility to or so far latent type 2 diabetes.

The current study was designed to test our research hypothesis, postulating the following questions to be explored in the current study: (i) MS in women with GDM in pregnancy identifies a patient subgroup with type 2 diabetes and the subsequent high risk of persistence of glucose metabolism abnormality postpartum and (ii) the presence of MS represents an increased risk of peripartal adverse outcomes for offspring compared with those of non-MS GDM women.

The aims of this study were (1) to ascertain the prevalence of MS in patients with GDM at the time of its diagnosis (mid-gestation) using modified criteria for MS (best fit for pregnancy status), (2) to ascertain a prevalence of MS in the subgroup of GDM patients with persistence of impaired glucose tolerance after delivery, and (3) to assess whether GDM patients with MS have a higher risk of peripartal adverse outcomes.

## 2. Materials and Methods

### 2.1. Subjects

This cross-sectional observational study included a total of 455 GDM participants (all Caucasians of Czech nationality from the South Moravian Region, Czech Republic) enrolled between 2013 and 2019. All consenting women diagnosed with GDM were included in this study during the enrolment period. The inclusion criteria were positive screening for GDM by oGTT at mid-gestation (for details, see the subsequent Section 2.2—Methods), singleton pregnancy, and Caucasian origin. The exclusion criteria were pre-existing type 1 diabetes or type 2 diabetes with established treatment before pregnancy (diagnosed according to recent WHO criteria [8]), non-Caucasian origin, and multiple pregnancies. All participants were followed from the time of GDM diagnosis to delivery at the Diabetes Centre of the Faculty Hospital Brno. This study was approved by the Ethical Committee of the Faculty of Medicine, Masaryk University, Brno, Czech Republic, and was conducted in accordance with the Declaration of Helsinki as revised in 2008. Each participant provided informed consent prior to being included in this study. At the time of GDM diagnosis at mid-gestation, other selected (co)morbidities were classified incl. hypothyroidism, allergies, polycystic ovary syndrome, preeclampsia, and anaemia. The prevalence of comorbidities on top of GDM diagnosis in the subset of subjects with and without GDM is shown in Table 2.

Of the total study sample (*n* = 455) 65% of subjects (*n* = 295) delivered in Obstetrics and Gynaecology Clinics of the University Hospital Brno, and peripartal data were therefore available to be retrieved retrospectively (2015–2021) by investigators from electronic health records. The analysed peripartal parameters comprised data on (i) the length of delivery (≥480 min duration of all three stages of labour counted defined as prolonged), (ii) the mode of delivery (the need for delivery induction, instrumental delivery, or Caesarean section), (iii) delivery complications (such as manual extraction of placenta or uterine hypotonia) and, finally, (iv) the selected neonatal parameters (Apgar score, the pH of cord blood, base excess (BE), and child birth weight). A list of participating units of the University Hospital Brno is available in Appendix A.

### 2.2. Methods

All enrolled subjects underwent fasting plasma glucose (FPG) testing in the first trimester of pregnancy by their gynaecologists, and all of them had normal FPG values. Subsequently, all subjects underwent routine mid-gestational GDM screening by the oral glucose tolerance test (oGTT) with 75 g of glucose the between 24 and 28th weeks of pregnancy (mid-trimester). GDM was diagnosed according to the IADPSG criteria (FPG ≥ 5.1 mmol/L, 1-h post-load glucose ≥ 10.0 mmol/L, and 2-h post-load glucose ≥ 8.5 mmol/L with any one of the three cut-off values qualifying for a GDM diagnosis). The following parameters were considered for analysis: age at the time of GDM diagnosis, pre-gestational BMI, systolic blood pressure (SBP), and diastolic blood pressure (DBP) in the second trimester of pregnancy. Triacylglycerols (TAGs), total cholesterol (TC), low-density lipoprotein cholesterol (LDL), and high-density lipoprotein cholesterol (HDL) were determined additionally from the samples of archived plasma of peripheral venous blood taken from each participant between the 24th and 30th week of pregnancy (by a diabetologist during their first visit to the Diabetes Centre). Plasma was separated by centrifugation (12,850 g, 10 min, 4 °C) and stored at −70 °C until analysis. Automatic analysis was performed using the clinical chemistry module c702 of the Cobas 8000 (F. Hoffman-La Roche Ltd.; hereinafter, Basel, Switzerland) via photometric tests (TAG Ref.05171407 190, total cholesterol Ref.05168568 190, LDL-cholesterol Ref.07005768 190, and HDL-cholesterol Ref.07528582 190).

All enrolled subjects were invited to participate in the postpartum oGTT, and 48% (*n* = 219) GDM patients underwent repeated oGTT tests up to 1 year after delivery with 11.4% (*n* = 25) manifesting permanent glucose intolerance (diabetes or prediabetes, evaluated according to WHO criteria for non-pregnant subjects: FPG ≥ 7 mmol/L alone or 2-h post-load glucose ≥ 11.1 mmol/L for diabetes mellitus or 5.6–6.9 mmol/L or 7.8–11.0 mmol/L for, respectively, prediabetes). In the case of a positive postpartum test, urinary ketone bodies, C-peptide, and the selected antibodies (anti-glutamic acid decarboxylase, anti-tyrosine phosphatase–2, and insulin autoantibodies) were measured to exclude eventual type 1 diabetes.

### 2.3. Definition of MS in the 2nd Trimester of Pregnancy

The International Diabetes Federation (IDF) [9] and WHO [10] criteria for MS definition were modified to the pregnancy situation, and the presence of a minimum of 3 of the 5 criteria was required for diagnosis of MS: GDM (as an obligatory criterion) + at least 2 of the following: BMI before pregnancy ≥30 kg/m^2^, blood pressure (BP) > 130/85 mmHg, TAG > 1.7 mmol/L, HDL < 1.3 mmol/L. With the exception of BMI, all parameters were evaluated in the second trimester of pregnancy; for more details, see Table 1. Of the two diagnostic criteria, we used a lower cut-off for BP and a higher cut-off for HDL (from IDF definition) considering physiological changes in these parameters in pregnancy. It was shown that during physiological pregnancy, BP decreases in the first half of pregnancy (under the influence of progesterone and prostaglandins, and due to increased uteroplacental circulation) up to the lowest values around mid-gestation. Later, BP increases to pre-gravid values [11,12]. Regarding the lipids—TC, TAGs, and HDL levels gradually increase throughout pregnancy. Lipids are low in early pregnancy, and then an accumulation of maternal fat depots is followed by increased adipose tissue lipolysis and subsequent hyperlipidaemia in late pregnancy (highest in the 2nd trimester). The LDL/HDL ratio is fairly stable [13,14].

### 2.4. Statistical Analysis

The data were expressed as medians and interquartile ranges (IQRs) or percentages for between-group comparisons. The Shapiro–Wilk normality test was used to test for a normal distribution. Nonparametric tests were used for comparison between and within the groups (Mann–Whitney and Wilcoxon tests, respectively). A chi-square test was used for contingency tables. Statistica (StatSoft, Tulsa, OK, USA) software was used for all analyses. *p* < 0.05 was considered statistically significant.

Univariate logistic models were constructed to determine an eventual statistically significant effect of any relevant variable and receiver operating characteristic (ROC) analysis was applied to test the final models. Areas under the ROC curve (AUC_ROC_) were compared by the Delong paired test [15]. Univariate logistic regression was used for the analysis of the relationship between MS parameters and the prediction of postpartum GDM conversion into permanent glucose abnormality within 12 months. Optimal cut-offs were selected by the highest Youden indices [16], i.e., single statistic capturing diagnostic test performance (J = sensitivity + specificity − 1) with values ranging from 0 to 1 (a zero value for the test giving the same proportion of positive results for groups with and without the disease and a value of 1 for no false positives or negatives).

## 3. Results

### 3.1. Prevalence of MS in Patients at the Time of GDM Diagnosis

Fully developed MS was detected in 22.6% (*n* = 103) of GDM patients at the 24th–28th week of pregnancy using modified criteria for MS (see Table 1). The comparison of basic anthropometric, clinical, and biochemical data between groups with and without MS showed significantly higher FPG in the MS group, and, not surprisingly, all parameters comprised the criteria of MS. For details, see Table 2. Table 3 shows the frequency of diagnostic/above cut-off values of particular parameters defining MS in women with and without MS in the GDM study population. BMI and TAG above the cut-offs were the most frequent diagnostic criteria, while HDL played a marginal role. Three (of five) diagnostic criteria were positive in 80.5% of the women with MS, four diagnostic criteria in 15.5%, and all five criteria were present in 4% of the women (*n* = 103 = 100%),

### 3.2. Prevalence of MS in the Subgroup of GDM Patients with a Persistence of Impaired Glucose Tolerance after Delivery

In patients with any form of persistent glucose intolerance after delivery (*n* = 25), the prevalence of MS was as high as 40%. Table 4 shows a comparison of basic anthropometric, clinical, and biochemical data in the time of GDM diagnosis between groups with and without impaired glucose tolerance (based on the results of postpartum oGTT up to 1 year after delivery). A prevalence of persistent glucose intolerance after delivery did not correlate with an increasing number of MS criteria.

Univariate logistic models were constructed to determine an eventual statistically significant effect of particular parameters contributing to MS in the prediction of the risk of postpartum persistence of glucose intolerance. ROC analysis was performed, and the optimal cut-offs were selected by the highest Youden indices. Table 5 shows fasting glycaemia with a cut-off above 5.6 mmol/L as the highest risk for the persistence of glucose intolerance postpartum (OR = 4.52, CI 2.38–8.57, *p* = 4 × 10^−6^), a BP above 135/85 mmHg (both values) was also significant (OR = 2.63, CI 1.18–5.82, *p* = 0.018). Surprisingly, BMI was a significant predictor when expressed as tertiles with the second BMI tertile associated with the minimal risk of conversion (used as reference), while the first tertile and the third tertile had increased risk compared with the second tertile (OR = 3.58, CI 1.13–11.34, *p* = 0.03 for the first tertile, OR = 3.97, CI 1.27–12.48, *p* = 0.018 for the third tertile, resp.). TAG and HDL levels were not found to be significant for the prediction of postpartum GDM conversion into permanent glucose abnormality, as well as first- and second-hour glycaemia in the oGTT test.

### 3.3. Risk of Peripartal Adverse Outcomes

The presence of MS in GDM patients was statistically significantly associated with two adverse peripartal outcomes: higher incidence of pathologic Apgar score and macrosomia (*p* = 0.01 and *p* = 0.0004, respectively, chi-square test). For details, see Table 6. When analysing a birth weight in offspring with macrosomia, we detected a significant positive correlation between birth weight (as well as an occurrence of macrosomia per se) and an increasing number of MS criteria (*p* < 0.05, r = 0.36 and r = 0.49, resp., Spearman). We also detected a positive correlation between birth weight and TC and LDL levels of mothers in the second trimester of pregnancy (*p* < 0.05, r = 0.34 and r = 0.40, resp., Spearman), while for TAG and HDL levels, no such correlations were ascertained. Moreover, women with children with macrosomia had significantly higher total cholesterol levels (*p* = 0.04, Mann–Whitney test) in the second trimester of pregnancy. Other lipids measured in the second trimester of pregnancy did not exhibit any influence on macrosomia.

## 4. Discussion

GDM is the most common complication of pregnancy that should—by definition—regress after delivery. Nevertheless, there is a substantial clinical heterogeneity in GDM with a subset of pregnant GDM women whose glucose intolerance persists postpartum. The exact proportion is not known precisely enough due to generally low compliance with postpartum testing in real-world settings. This subset of GDM cases might suffer from the first occurrence of type 2 diabetes unmasked by pregnancy that—owing to its persistence after delivery—puts the women at especially high risk of vascular and other diabetes-related complications. The identification of these high-risk subjects can improve the management of glucose metabolism abnormality and prevent complication development with a corresponding reduction in health care costs.

Even a completely physiological pregnancy is marked by a significant decrease in insulin sensitivity around mid-gestation as a consequence of fully matured placenta-derived hormones counter-balancing insulin action. Pre-existing pathological baseline insulin resistance in women with well-identified GDM risk factors such as an advancing age of motherhood, overweight/obesity, and some other less well-defined alterations incl. possible genetic contribution then superimposes and further exacerbates the physiological one and, in combination with beta-cell dysfunction (identical to type 2 diabetes), results in the development of clinically manifest GDM. We designed the current study to test our hypothesis that MS detectable at the same time as GDM in mid-gestation can identify subjects with an increased risk of glucose abnormality persistence postpartum. Our major findings can be summarised as follows: using our modified criteria, MS was detected in 22.6% of women with GDM in the second trimester of pregnancy. Moreover, in a subgroup of patients with any form of persistent glucose intolerance after delivery, the prevalence of MS reached as high as 40%. Furthermore, FPG above 5.6 mmol/L together with a BP above 135/85 increased the risk of persistence of postpartum glucose intolerance. Interestingly, the BMI ROC curve derived by univariate logistic models revealed bidirectional risk when considering BMI tertiles. The first tertile (leanness to undernourishment values) and the third tertile (with values close to obesity) were found to be a statistically significant risk for postpartum persistence of glucose abnormalities. We can speculate, that women from those two different groups could have a different pathophysiology of glucose abnormality development in pregnancy. Since a rigorous assessment of insulin sensitivity and secretion was not performed, we can only speculate on the reasons for the contra-intuitively increased risk in the first tertile compared with the second. One possibility is the manifestation of the non-T1DM/non-T2DM primary type of diabetes in pregnancy such as MODY. Definitely, more studies are needed to elucidate this phenomenon since very few studies are aimed at this topic [17,18]. Finally, increased levels of lipids during the second trimester of pregnancy (TC and LDL) play a role in the occurrence of macrosomia, and our findings are therefore in accordance with other published studies [19,20]. Only a few studies found a significantly increased TAG in GDM [21], but studies have not specifically focused on postpartum persistence of glucose intolerance. Last but not least, the coincidence of MS and GDM increases the risk of a pathological Apgar score during delivery.

MS was more commonly studied before pregnancy or early in the first trimester [22,23,24] as well as after pregnancy using standard diagnostic criteria [25]. The study of 5530 low-risk, nulliparous women recruited from the multi-centre, international prospective Screening for Pregnancy Endpoints (SCOPE) study [22] found that more than half of the women who had MS in early pregnancy developed a pregnancy complication compared with just over a third of women who did not have MS. Furthermore, while an increasing BMI increases the probability of GDM, the addition of MS exacerbates this probability. In a Saudi study comprising 498 pregnant women [23] that also aimed at the first trimester, MS prevalence was found to be significantly higher in GDM participants (25%); moreover, those with later-diagnosed GDM had hyperglycaemia and hypertriglyceridemia in the first trimester. A review and meta-analysis of 23 studies [25] (10,230 pregnant women: 5169 cases and 5061 controls) indicated that women with a history of GDM had a higher risk of developing MS compared with those without such a history, and the risk of developing MS was even higher in studies where women with GDM had an increased BMI compared with the controls.

On the contrary, studies of MS during later pregnancy—ideally, at the time of routine screening for GDM—are scarce. Yet, the established and widely administered screening of GDM in the second trimester (24th–28th week of gravidity) offers so far not an entirely harvested opportunity to manage metabolic and cardiovascular risks more effectively. Pregnancy is a short period in the life of women when, due to hormonal and nutritional influences, so far latent metabolic derangements can be unmasked, and they can indicate not only immediate risks (for the remaining pregnancy duration and delivery) but also lifelong risks operating potentially early after delivery. Understanding the factors determining the degree of postpartum metabolic impairment would allow for better education and early management.

The limitations of our study have to be mentioned. Firstly, there are no established diagnostic criteria for MS in the second trimester of pregnancy. We, therefore, adjusted the current criteria according to available physiological data on cardiovascular and metabolic parameters in mid-gestation. Yet, more data are needed to eventually improve the diagnostic precision of MS during pregnancy and increase specificity. Secondly, the disparate numbers of women with a GDM history who normalised glucose metabolism vs. those with persistent glucose metabolism impairment might affect the robustness of the findings and our ability to reveal possible pathophysiological differences in the origin of GDM among specific subgroups of pregnant women. We are therefore currently enrolling an independent cohort of GDM subjects followed postpartum in order to replicate our findings.

## 5. Conclusions

In conclusion, the presence of MS in GDM patients represents a statistically significant risk factor (*p* = 0.04, chi-square test) for the persistence of glucose intolerance after delivery that also negatively influences the selected outcomes of delivery. MS could be diagnosed together with GDM during routine screening in the second trimester of pregnancy. Women with a higher risk of adverse peripartal outcomes together with the risk of persistence of glucose abnormalities after delivery can be thus identified early and effective lifestyle education and management could be administered effectively.

## Figures and Tables

**Table 1 jcm-13-01260-t001:** Criteria for MS definition.

IDF Definition of MS 2020 (Women)	WHO Definition of MS 2020 (Women)	Current Study Modified Definition of MS in the Second Trimester
Waist circumference ≥ 80 cm + at least two criteria from:	Any form of insulin resistance (FPG ≥ 5.6 mmol/L or IGT ≥ 7.8 mmol/L or DM) + at least two criteria from:	GDM (24/28th week of gravidity) according to IADPSG + at least two criteria from:
TAG ≥ 1.7 mmol/L (or therapy)	TAG ≥ 1.7 mmol/L (or therapy)	TAG ≥ 1.7 mmol/L *
HDL-C ≤ 1.3 mmol/L (or therapy)	HDL-C ≤ 1.0 mmol/L	HDL ≤ 1.3 mmol/L *
BP ≥ 130/85 mmHg (or therapy)	BP ≥ 140/90 mmHg (or therapy)	BP ≥ 130/85 mmHg *
FPG ≥ 5.6 mmol/L or DM	BMI ≥ 30 kg/m^2^ or waist/hip ratio ≥ 0.85	BMI before pregnancy ≥30 kg/m^2^

* measured in the second trimester of gravidity. BMI—body mass index, BP—blood pressure, DM—diabetes mellitus, FPG—fasting plasma glucose, GDM—gestational diabetes mellitus, HDL—high-density lipoprotein, IADPSG—International Association Diabetes Pregnancy Study Group, IDF—International Diabetes Federation, MS—metabolic syndrome, TAG—triacylglycerol, WHO—World Health Organisation.

**Table 2 jcm-13-01260-t002:** Comparison of clinical and biochemical parameters of GDM women with MS and without MS.

Parameter	Without MS in Pregnancy (*n* = 352)	MS in Pregnancy (*n* = 103)	*p*
Age (years)	33 (30–36)	33 (29–36)	NS
BMI before pregnancy	25.0 (22.0–28.5)	32.7 (29.1–36.0)	<1 × 10^−6^
FPG 2nd trim. (mmol/L)	5.0 (4.6–5.4)	5.3 (4.7–5.7)	0.0006
oGTT 2nd h 2nd trim. (mmol/L)	8.7 (8.0–9.3)	8.4 (7.4–9.4)	NS
SBP 2nd trim. (mmHg)	115 (107–122)	131 (123–138)	<1 × 10^−6^
DBP 2nd trim. (mmHg)	74 (68–80)	82 (76–90)	<1 × 10^−6^
HDL 2nd trim. (mmol/L)	1.8 (1.6–2.0)	1.6 (1.2–1.8)	0.001
TAG 2nd trim. (mmol/L)	2.1 (1.5–2.5)	2.4 (2.0–3.1)	0.003
TC 2nd trim. (mmol/L)	6.2 (5.3–7.1)	5.8 (5.3–6.4)	NS
Comorbidites (%)	49.4	57.3	NS
Offspring birth weight (g)	3465 (3155–3745)	3500 (3250–3950)	NS

Data are expressed as median (IQR), Mann–Whitney test, or % (chi-square test). BMI—body mass index, DBP—diastolic blood pressure, FPG—fasting plasma glucose, GDM—gestational diabetes mellitus, HDL—high-density lipoprotein, MS—metabolic syndrome, oGTT—oral glucose tolerance test, SBP—systolic blood pressure, TAG—triacylglycerol, TC—total cholesterol, comorbidities—presence at least one other of the selected diagnoses on top of GDM.

**Table 3 jcm-13-01260-t003:** Frequency of MS diagnostic parameters in women with and without MS in the GDM study population (i.e., on top of obligatory GDM criterion).

Without MS in Pregnancy (*n* = 352)		MS in Pregnancy (*n* = 103)	
BP	10.5%	TK	52%
BMI	15.3%	BMI	74%
TAG	21.9%	TAG	70%
HDL	1.4%	HDL	30%

Data are expressed as % above the cut-offs of the selected parameters. BMI—body mass index, BP—blood pressure, DM—diabetes mellitus, GDM—gestational diabetes mellitus, HDL—high-density lipoprotein, MS—metabolic syndrome, TAG—triacylglycerol.

**Table 4 jcm-13-01260-t004:** Comparison of clinical and biochemical parameters of GDM women with normal or abnormal oGTT after delivery.

Parameter	Normal oGTT after Delivery (*n* = 194)	Prediabetes/Diabetes after Delivery (*n* = 25)	*p*
Age (years)	32 (30–36)	30 (28–35)	NS
BMI before gravidity	25.2 (22.5–29.1)	28.4 (20.8–32.7)	NS
FPG second trim. (mmol/L)	4.9 (4.6–5.3)	5.7 (5.2–6.2)	5 × 10^−6^
oGTT second h second trim. (mmol/L)	8.7 (7.8–9.3)	9.3 (8.0–10.4)	0.04
SBP second trim. (mmHg)	117 (109–127)	124 (113–130)	NS
DBP second trim. (mmHg)	74 (69–81)	80 (70–87)	NS
HDL second trim. (mmol/L)	1.8 (1.6–2.0)	1.5 (1.2–1.8)	NS
TC second trim. (mmol/L)	6.0 (5.3–6.7)	5.8 (5.5–6.5)	NS
TAG second trim. (mmol/L)	2.1 (1.7–2.6)	2.9 (2.0–3.1)	NS
MS (%)	18.6%	40.0%	0.04
Birth weight (g)	3420 (3050–3660)	3520 (3500–3950)	NS

Data are expressed as median (IQR), Mann–Whitney test, MS as frequency (%), and chi-square test. BMI—body mass index, DBP—diastolic blood pressure, DM—diabetes mellitus, FPG—fasting plasma glucose, GDM—gestational diabetes mellitus, HDL—high-density lipoprotein, MS—metabolic syndrome, oGTT—oral glucose tolerance test, SBP—systolic blood pressure, TAG—triacylglycerol.

**Table 5 jcm-13-01260-t005:** ROC analysis for particular parameters concluded in metabolic syndrome for the prediction of postpartum GDM conversion into permanent glucose abnormality. The cut-offs show a value with the highest sensitivity/specificity.

Parameter	Cut Off According to the Youden Index ^a^		AUC_ROC_	ODDs Ratio	95% CI ^b^	*p*
Fasting glycaemia	>5.6 mmol/L		0.77	4.52	2.38–8.57	4 × 10^−6^
BP	>135/85 mmHg		0.61	2.63	1.18–5.82	0.018
TAG	>2.69 mmol/L		0.72	1.84	0.65–5.61	0.28 (NS)
HDL	<1.65 mmol/L		0.68	1.47	0.33–6.60	0.62 (NS)
BMI	Second tercile at minimum significant for conversion BMI 22.68–27.28	First tercile vs. second tercile	0.62	3.58	1.13–11.34	0.030
		Second tercile vs. third tercile	0.62	3.97	1.27–12.48	0.018

^a^ Youden index (Youden, 1950) (i.e., sensitivity+ specificity-1), ^b^ Binomial exact (binomic exact test for the calculation of confidence intervals (CIs)). ROC—receiver operating characteristic; AUC_ROC_—area under the ROC curve; CI—confidence interval; BMI—body mass index; AUC—area under the curve, BP—blood pressure, HDL—high-density lipoprotein, TAG—triacylglycerol.

**Table 6 jcm-13-01260-t006:** Comparison of “peripartal adverse outcomes“ in GDM women with and without MS.

Parameter	Without MS in Pregnancy (*n* = 232)	MS in Pregnancy (*n* = 63)	*p*
Induction of delivery	40.1%	36.0%	NS
Instrumental delivery (section, vacuum extractor, obstetric pliers)	31.5%	41.3%	NS
Protracted delivery (more than 480 min)	23.3%	31.7%	NS
Macrosomia (birth weight above 4000 g)	5.2%	19.0%	0.0004
Abnormal Apgar score (in fifth min <5)	1.3%	6.3%	0.01

Data are expressed as frequency, chi-square test. Abnormalities in base excess and pH of cord blood were not detected in the study group.

## Data Availability

The data are available upon request from the authors.

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
