# Peer review of "Metabolic Syndrome Prevalence in Women with Gestational Diabetes Mellitus in the Second Trimester of Gravidity"

_jcm, 2024, doi:10.3390/jcm13051260_

Round 1
Reviewer 1 Report
Comments and Suggestions for Authors
1. Please cited the original article for the definition of prolonged labor greater and equal to 8 hours.
2. Please provide more explanation for the increase risk of postpartum persistent of glucose intolerance in first tertile cohort.
Author Response
- Please cited the original article for the definition of prolonged labor greater and equal to 8 hours.
- Thank you for spotting this lack of definition (although we used this cut-off before in other papers without rising concern). Nonetheless, we agree with the reviewer this deserves attention. Definition of prolonged labour is inherently difficult and it is a controversial issue (ad reviewed elsewhere – see below) since the definition might be context dependent. Different health care practices (spontaneous vs. induced labour, guidelines for administration of oxytocin etc.), parity, age etc. are potentially confounders of the absolute definition of prolonged labour.
- Nystedt A, Hildingsson I. Diverse definitions of prolonged labour and its consequences with sometimes subsequent inappropriate treatment. BMC Pregnancy Childbirth. 2014 Jul 16;14:233. doi: 10.1186/1471-2393-14-233. PMID: 25031035; PMCID: PMC4105110.
- The Czech Gynaecology Society defines prolonged delivery defined as delivery with slow progression- whether in the first or second stage – when the baby isn't born after 18 or more hours of contractions. Internal consensus in the participating centre for high-risk women incl. GDM is stricter in the time range and defines 8 hrs as the upper threshold for normally proceeding delivery.
- Should the reviewer feel this full explanation is necessary in the MS, we are more than happy to add this text into the respective section of M a M.
- Thank you for spotting this lack of definition (although we used this cut-off before in other papers without rising concern). Nonetheless, we agree with the reviewer this deserves attention. Definition of prolonged labour is inherently difficult and it is a controversial issue (ad reviewed elsewhere – see below) since the definition might be context dependent. Different health care practices (spontaneous vs. induced labour, guidelines for administration of oxytocin etc.), parity, age etc. are potentially confounders of the absolute definition of prolonged labour.
- Please provide more explanation for the increase risk of postpartum persistent of glucose intolerance in first tertile cohort.
- We are indeed intrigued by this result and we thank reviewer for the deep interest in this finding. We extended discussion of this finding in the respective section of Discussion in p.8-9.
Reviewer 2 Report
Comments and Suggestions for Authors
Dear Author,
This is a valuable article showing that the presence of an additional diagnosis of metabolic syndrome is associated with worse perinatal outcomes in patients diagnosed with gestational diabetes. A physiological increase in total cholesterol and triglyceride levels is observed during pregnancy. In this article, the diagnostic criteria for metabolic syndrome determined by different societies are discussed. However, I have some suggestions below.
Abstract- Line 18: Add ‘International Diabetes Federation’ prior to IDF
The information in the introduction section can be summarised appropriately.
The methodology has been meticulously designed and effectively implemented.
The results are presented clearly and concisely.
Relevant references are appropriately discussed in the discussion section.
A native English speaker should revise the manuscript for clarity.
The topic is interesting enough to attract the readers’ attention.
Comments on the Quality of English LanguageEnglish language editing is required.
Author Response
Thank you for a very pleasant evaluation, we incorporate comments as you suggested
Abstract- Line 18: Add ‘International Diabetes Federation’ prior to IDF – thank you, we added the information
A native English speaker should revise the manuscript for clarity – indeed, native speaker was asked to revise MS and changes were implemented when appropriate
Reviewer 3 Report
Comments and Suggestions for Authors
The authors performed a clinical study on GDM and metabolic syndrome. This is a comparatively delicate study with some interesting data. Here are the comments from the reviewer:
1. All tables should be in three lines.
2. Please list the centers in this study as a supplement table.
3. The baseline, especially the basic diseases of these patients, should be provided.
4. TG, TC levels should be included in this study.
5. The authors should supplement a figure about how many patients were included in this study, how many patients were excluded, and how many patients withdrawal from this study.
6. Please clearly list the inclusion and exclusion criteria in the method section.
Comments on the Quality of English LanguageEnglish language is fine.
Author Response
The authors performed a clinical study on GDM and metabolic syndrome. This is a comparatively delicate study with some interesting data. Here are the comments from the reviewer:
- All tables should be in three lines.
- Thanks you for your suggestion. However, we kindly leave the decision on the table format on editor/graphical editor. Subsequently, we are ready to make any changes recommended.
- Please list the centers in this study as a supplement table.
- Thank you for this suggestion. Given the rather limited number of participating centres – two clinics of University Hospital Brno (Diabetes unit of Internal clinic and Obstetric and Gynaecology clinic) we have not felt the need. But the Supplementary table 1 is now prepared and was added in the respective section.
- The baseline, especially the basic diseases of these patients, should be provided.
- Thank you, this is a very good point, we added the information at Material and Methods, section 2.1 and prevalence of comorbidities is shown currently also in Table 2.
- TG, TC levels should be included in this study.
- Triacylglycerol levels (denoted TAG) are included in the study – shown in Table 1 – 6.
- Total cholesterol (TC) was also measured and data are available. TC was not shown in the original version of the MS since this parameter does not add to the MS definition. However, we now added this parameter in the Table 2 showing TC levels together with other variables in subjects with and without MS.
- The authors should supplement a figure about how many patients were included in this study, how many patients were excluded, and how many patients withdrawal from this study.
- During the enrolment we did not carried a specific record of how many patients were not eligible (either due to not meeting inclusion criteria or not giving informed consent). Such information does not routinely appear in clinical studies.
- Regarding the withdrawal, there was not specific case of particular patient withdrawing voluntarily from the study.
- M and M, section 2.1 explicitly specifies how many patients were diagnosed with GDM in the University hospital (therefore available for analysis of MS prevalence in pregnancy) and how many of them delivered in the same facility (therefore available for the analysis of obstetric and postpartum data).
- Please clearly list the inclusion and exclusion criteria in the method section.
- Thank you very much for this suggestion. The respective criteria were clearly defined in M and M section 2.1.
Round 2
Reviewer 3 Report
Comments and Suggestions for Authors
The authors carefully revised the manuscript. I would recommend acceptance of this manuscript with extensive English language editing.
Comments on the Quality of English LanguageExtensively language editing is required.